# Physics-Informed Neural Networks for Solving Coupled Stokes–Darcy Equation

**DOI:** 10.3390/e24081106

**Published:** 2022-08-11

**Authors:** Ruilong Pu, Xinlong Feng

**Affiliations:** College of Mathematics and System Sciences, Xinjiang University, Urumqi 830017, China

**Keywords:** Stokes–Darcy equation, interface conditions, deep learning method, deep neural network, physics-informed neural network

## Abstract

In this paper, a grid-free deep learning method based on a physics-informed neural network is proposed for solving coupled Stokes–Darcy equations with Bever–Joseph–Saffman interface conditions. This method has the advantage of avoiding grid generation and can greatly reduce the amount of computation when solving complex problems. Although original physical neural network algorithms have been used to solve many differential equations, we find that the direct use of physical neural networks to solve coupled Stokes–Darcy equations does not provide accurate solutions in some cases, such as rigid terms due to small parameters and interface discontinuity problems. In order to improve the approximation ability of a physics-informed neural network, we propose a loss-function-weighted function strategy, a parallel network structure strategy, and a local adaptive activation function strategy. In addition, the physical information neural network with an added strategy provides inspiration for solving other more complicated problems of multi-physical field coupling. Finally, the effectiveness of the proposed strategy is verified by numerical experiments.

## 1. Introduction

The coupled Stokes–Darcy equation studied in this paper has many applications in the physical and engineering sciences. For example, in reservoir modeling, to model heterogeneous porous media, the permeability field is often assumed to be a multiscale function with high contrast and discontinuous features. There are also model studies of the evolution of fibroblast shape and position under stress [1]. The model is based on the idea of continuum mechanics to describe the stress-induced phase transition, the cell body is modeled as a linear elastic matrix, and the cell body surface evolves according to a specific dynamic relationship. In this model, the stress tensor has discontinuities at the cell surface due to changes in the strain tensor due to cell contraction. The stiffness term in the Stokes–Darcy system due to small parameters and the discontinuity in the normal velocity due to the imbalance of the normal stress at the interface make our problem difficult to solve. Moreover, for complex area problems, curved interface problems, and high-dimensional problems, it will be difficult to mesh generation. Therefore, knowing how to design an accurate, efficient, and stable meshless numerical approximation algorithm has become the focus of literature research. Studies [2,3,4,5,6,7] are relevant here, and interested readers can read the research.

In recent years, deep learning methods have achieved unprecedented success in various application fields such as computer vision, speech recognition, natural language processing, audio recognition, social network filtering, and bioinformatics. In some cases, they are better than human experts [8,9]. Driven by these exciting developments, people began to make an in-depth study on how to use deep neural networks to solve partial differential equations [10,11,12,13]. In particular, Raissi et al. proposed physics-informed neural networks(PINNs) to help solve partial differential equations and data-driven discovery [14]. The results show that given certain initial conditions and boundary conditions, PINNs can solve some partial differential equations very well. Since then, the door to solve partial differential equations using deep neural networks has been opened, and some works [15,16,17,18,19,20,21,22,23,24] based on PINNs have been published one after another. For example, Even Lu Lu et al. expounded the difference between the traditional finite element method and the deep neural network in solving partial differential equations from the selection of basis functions, the solution process, the error source, and the error order in [25]. Even the famous Mishra [26] et al. and Jagtap [27,28] et al. made a theoretical analysis on the errors generated in the training process of PINNs, in their respective works. Mishra et al. conducted a generalization error analysis to solve a class of inverse problems of PINNs, and Jagtap et al. carried out an error analysis on PINNs approaching the Navier–Stokes equation and generalized an error of extended physics-informed neural networks (XPINNs).

This paper mainly wants to study a new method for the numerical approximation of a meshless deep neural network [29] to solve the problems we care about. Our main goal is to study strategies to improve the ability of deep neural network to solve the Stokes–Darcy model, and to improve the approximation ability of PINNs to solve the Stokes–Darcy model. We propose strategies to improve the accuracy and efficiency of deep neural networks in this paper and provide several numerical examples to demonstrate our approach. It should be noted that due to the randomness of the initial parameters when training the network, our numerical results will fluctuate within a certain range.

The structure of this paper is as follows: an introduction to the Stokes–Darcy fluid coupling problem and its mathematical model is given in Section 2. In Section 3, the related knowledge of the neural network and strategies to improve the approximation properties of PINNs are introduced. The accuracy and reliability of the proposed strategy are verified by numerical examples in Section 4. Concluding remarks and outlook are given in Section 5.

## 2. Problem Setup

In this part, we specifically introduce the mathematical model of the problem and the corresponding interface conditions. The Stokes–Darcy fluid coupling system is discussed in a given region Ω; Γ divides region Ω into $\Omega_{s}$ and $\Omega_{d}$, representing the region of Stokes flow and Darcy flow, respectively. For simplicity, $\Gamma_{s}$ and $\Gamma_{d}$ represent the boundaries of $\Omega_{s}$ and $\Omega_{d}$, except the interface. n and τ are used to represent the external normal vector and tangent vector respectively. On interface Γ, $n_{s}$ is used to represent the normal vector from region Stokes to region Darcy, $n_{d}$ is used to represent the normal vector from region Darcy to region Stokes, and τ is the tangent vector.

In order to better describe the Stokes–Darcy fluid-coupling system mathematically, first of all, the motion of fluids in $\Omega_{s}$ and $\Omega_{d}$ is described by the Stokes equation and Darcy law, respectively. We often need to distinguish between physical quantities in $\Omega_{s}$ and $\Omega_{d}$, especially when they are at the interface Γ. Therefore, the relevant physical quantities in the $\Omega_{s}$ region are represented by the symbols with the subscript s, and the relevant physical quantities in the $\Omega_{d}$ region are represented by the symbols with the subscript d, as follows:$$u_{s}{=u|}_{\Omega_{s}},u_{d}{=u|}_{\Omega_{d}},p_{s}=p{{|}_{\Omega_{s}},p_{d}=p|}_{\Omega_{d}}.$$

So we can get the governing system [7,30,31] as follows:

**Fluid region** (Stokes equations)
(1)$$\begin{array}{cccc} -\nabla \cdot T(u_{s},p_{s})= & f_{s}, & \mathrm{in} & \;\Omega_{s},\\ \nabla \cdot u_{s}= & 0, & \mathrm{in} & \;\Omega_{s}. \end{array}$$
where $u_{s}$ is the fluid velocity, $p_{s}$ is the kinematic pressure, $f_{s}$ is the external force, ν>0 is the kinematic viscosity of the fluid, $T\left(u_{s},p_{s}\right)=2\nu D\left(u_{s}\right)-p_{s}I$ is the stress tensor, and $D\left(u_{s}\right)=\frac{1}{2}\left(\nabla u_{s}+\nabla u_{s}^{T}\right)$ is the deformation tensor, and I is the unit vector.

**Rock matrix** (Darcy equations)
(2)$$\begin{array}{cccc} \nu K^{-1}u_{d}+\nabla p_{d}= & f_{d}, & \mathrm{in} & \;\Omega_{d},\\ \nabla \cdot u_{d}= & 0, & \mathrm{in} & \;\Omega_{d}. \end{array}$$

The above equation is the Darcy equation describing fluid flow in the porous media region, see [2,30,31,32], $u_{d}$ is the fluid velocity in the porous medium, $p_{d}$ is the dynamic pressure, and $f_{d}$ is the external force source term. The permeability K is a positive definite symmetric tensor allowed to vary in space.


**Outer boundary**

(3)
$$\begin{array}{cccc} u_{s}= & 0, & \mathrm{on} & \;\Gamma_{s},\\ u_{d}\cdot n_{s}= & 0, & \mathrm{on} & \;\Gamma_{d}. \end{array}$$



Here, for simplicity, we consider the Dirichlet boundary conditions on the Stokes side and the Neumann boundary conditions on the Darcy side.

Obviously, the pressure is unique under an additional constant, so we can assume that
$$\int_{\Omega }p\;\mathrm{dxdy}=0.$$


**Interface conditions**

(4a)
$$\begin{array}{cccc} u_{s}\cdot n_{s}+u_{d}\cdot n_{d}= & 0,\; & \mathrm{on} & \;\Gamma , \end{array}$$


(4b)
$$\begin{array}{cccc} 2\nu n_{s}\cdot D\left(u_{s}\right)\cdot n_{s}= & p_{s}-p_{d},\; & \mathrm{on} & \;\Gamma , \end{array}$$


(4c)
$$\begin{array}{cccc} 2n_{s}\cdot D\left(u_{s}\right)\cdot \tau = & -\alpha K^{-1/2}u_{s}\cdot \tau ,\; & \mathrm{on} & \;\Gamma . \end{array}$$



Here, the ([Disp-formula FD4a-entropy-24-01106]) is a continuity condition of normal velocity at an interface obtained by conservation of mass, ([Disp-formula FD4b-entropy-24-01106]) is a continuity condition of normal stress of fluid at an interface obtained through normal force balance, and ([Disp-formula FD4c-entropy-24-01106]) is a famous Beaver–Joseph–Saffman (BJS) interface condition [30,33,34,35], where parameter α is a constant associated with friction.

## 3. Numerical Method

In this part, we will adopt the fully connected deep neural network (DNN) as our basic network to solve the problem. At the same time, the PINNs algorithm framework and some extensions of the algorithm are introduced.

### 3.1. Network Formation

DNN is a widely parallel connected network composed of multiple simple units. Its organizational structure can simulate the interactive response of a biological nervous system to real-world objects. From the perspective of computer science, the neural network can be regarded as a mathematical model with multiple parameters. This is the result of nested functions, such as $y_{i}=f_{act}\left(\sum_{i}W_{i}x_{i}+b_{i}\right)$. We connect each neuron of each layer together. Taking the neural network of the L-layer as an example, the output of the neural network is as follows:(5)$$U\left(x,\theta \right)=W_{N_{L}-1}f_{act}\left(\cdots W_{2}f_{act}\left(W_{1}\left(x\right)+b_{1}\right)+b_{2}\cdots \right)+b_{N_{L}-1},$$
where $W_{i}$ is the weighting coefficient matrix and $b_{i}$ is the bias vector. All the undetermined parameters $\theta ={\left\{W_{i},b_{i}\right\}}_{i=1,2,\cdots ,N_{L}-1}\in \Theta$ in (Equation 5), and Θ is the parameter space. The (Equation 5) can also be written as
(6)$$U_{\theta }\left(z\right)=\left(N_{D}\circ \sigma \circ N_{D-1}\circ \sigma \circ N_{D-2}\circ \cdots \circ \sigma \circ N_{1}\right)\left(z\right).$$

Here, $N_{1}=W_{1}z+b_{1}$, z is the input variable of neural network, σ stands the activation function, and *D* represents the number of layers of the neural network.

### 3.2. Physics-Informed Neural Networks

In [14], the authors propose to use deep neural networks to approximate the solution of partial differential equations, which can be called u-networks, and then use automatic differential techniques to obtain the differential operators of the equation. They then obtain the f-network satisfying the physical information of the equation. Then, the boundary function and internal loss function are established by using the principle of least squares. The working process of the PINNs is better explained below by taking the model we are asking for as an example.

When solving the Stokes–Darcy equation, we use the random Latin hypercube random point method to extract the data points and divide the data points into five parts according to the problem. After the input of the neural network is determined, we need to use the given boundary conditions and equation information to establish the loss function. Generally, the least square method is used, and the automatic differentiation technology [36] is also used in this process. Here, we divide the loss function into five parts: $L(x_{fs},\theta )$ represents the internal loss of the Stokes region; $L(x_{fd},\theta )$ represents the internal loss of the Darcy region; $L(x_{u\Gamma },\theta )$ represents the loss on the interface; and $L(x_{us},\theta )$ and $L(x_{ud},\theta )$ represent the loss on the boundary of the Stokes region and the Darcy region, respectively. Additionally, the specific expressions are as follows:(7)$$L\left(x,\theta \right)=L\left(x_{fs},\theta \right)+L\left(x_{us},\theta \right)+L\left(x_{fd},\theta \right)+L\left(x_{ud},\theta \right)+L\left(x_{u\Gamma },\theta \right),$$
where,
$$\begin{array}{cc} L(x_{fs},\theta )= & \frac{1}{N_{fs}}\sum_{i=1}^{i=N_{fs}}\left[\right|-2\nu \nabla \cdot D\left(u_{s}\left(x_{f_{s}}^{i},y_{f_{s}}^{i}\right)\right)+\nabla p_{s}\left(x_{f_{s}}^{i},y_{f_{s}}^{i}\right)-f_{s}\left(x_{f_{s}}^{i},y_{f_{s}}^{i}\right){|}^{2}+|\nabla \cdot u_{s}\left(x_{f_{s}}^{i},y_{f_{s}}^{i}\right){|}^{2}],\\ L(x_{fd},\theta )= & \frac{1}{N_{fd}}\sum_{i=1}^{i=N_{fd}}[|\nu K^{-1}u_{d}\left(x_{f_{d}}^{i},y_{f_{d}}^{i}\right)+\nabla p_{d}\left(x_{f_{d}}^{i},y_{f_{d}}^{i}\right)-f_{d}\left(x_{i},y_{i}\right){|}^{2}+|\nabla \cdot u_{d}\left(x_{f_{d}}^{i},y_{f_{d}}^{i}\right){)|}^{2}],\\ L(x_{u\Gamma },\theta )= & \frac{1}{N_{u_{\Gamma }}}\sum_{i=1}^{i=N_{u_{\Gamma }}}\left[\right|u_{s}\left(x_{u_{\Gamma }}^{i},y_{u_{\Gamma }}^{i}\right)\cdot n_{s}+u_{d}\left(x_{u_{\Gamma }}^{i},y_{u_{\Gamma }}^{i}\right)\cdot n_{d}{|}^{2}+|2\nu n_{s}\cdot D\left(u_{s}\left(x_{u_{\Gamma }}^{i},y_{u_{\Gamma }}^{i}\right)\right)\cdot n_{s}\\ & -p_{s}\left(x_{u_{\Gamma }}^{i},y_{u_{\Gamma }}^{i}\right)+p_{d}\left(x_{u_{\Gamma }}^{i},y_{u_{\Gamma }}^{i}\right){|}^{2}+|2n_{s}\cdot D\left(u_{s}\left(x_{u_{\Gamma }}^{i},y_{u_{\Gamma }}^{i}\right)\right)\cdot \tau +\alpha K^{-1/2}u_{s}\left(x_{u_{\Gamma }}^{i},y_{u_{\Gamma }}^{i}\right)\cdot \tau {|}^{2}],\\ L(x_{us},\theta )= & \frac{1}{N_{us}}\sum_{i=1}^{i=N_{u_{s}}}{\left|u_{s}\left(x_{u_{s}}^{i},y_{u_{s}}^{i}\right)\right|}^{2},\\ L(x_{ud},\theta )= & \frac{1}{N_{ud}}\sum_{i=1}^{i=N_{ud}}{\left|u_{d}\left(x_{u_{d}}^{i},y_{u_{d}}^{i}\right)\cdot n_{s}\right|}^{2}. \end{array}$$

Here, ${\left\{x_{f_{s}}^{i},y_{f_{s}}^{i}\right\}}_{i=1}^{N_{f_{s}}}$ represents the configuration points inside the Stokes region; ${\left\{x_{f_{d}}^{i},y_{f_{d}}^{i}\right\}}_{i=1}^{N_{f_{d}}}$ represents the configuration points inside the Darcy region; ${\left\{x_{u_{\Gamma }}^{i},y_{u_{\Gamma }}^{i}\right\}}_{i=1}^{N_{u_{\Gamma }}}$ represents the training data on the interface; and ${\left\{x_{u_{s}}^{i},y_{u_{s}}^{i}\right\}}_{i=1}^{N_{u_{s}}}$ and ${\left\{x_{u_{d}}^{i},y_{u_{d}}^{i}\right\}}_{i=1}^{N_{u_{d}}}$ represent the training data on the $\Omega_{s}$ and $\Omega_{d}$ boundaries, respectively. $N_{fs}$, $N_{fd}$, $N_{u_{\Gamma }}$, $N_{us}$, and $N_{ud}$ represent the number of points in the Stokes region, the number of points in the Darcy region, the number of points in the interface, the number of points in the border of the Stokes region, and the number of points in the Darcy region. After establishing the loss function, we need to select the appropriate optimization algorithm to train the loss function and update the parameters in the neural network through training and back propagation. This process is repeated until the number of training sessions we set is reached or the loss function values converge. Then, we find an approximate solution of the partial differential equation. Common optimization algorithms include the stochastic gradient descent method, Newton method, and quasi-Newton method. This paper adopts the gradient based Adam algorithm [37], which has the advantages of adaptive learning rate and batch computing. In some calculation examples, the Adam algorithm is combined with the L-BFGS algorithm [38]. The working process of PINNs is given by Figure 1.

In Figure 1, x and y represent the input of the neural network; $f_{act}$ in (Equation 5) and σ in the figure both represent the activation function in the neural network; and u, v, and p represent the output of the neural network.

### 3.3. Improving Strategy of Physical-Informed Neural Network

The PINNs has a strong approximation ability, can solve many physical problems, and can describe many physical phenomena, but it has certain limitations in solving small parameter problems, such as the fluid viscosity coefficient and permeability in the Stokes–Darcy system. If the viscosity coefficient of the problem to be solved is very small, it will increase the difficulty of solving. At the same time, there are some limitations in solving the interface discontinuity problem. In the Stokes–Darcy equation, if the analytical solution is discontinuous on the interface, the general PINNs cannot be well solved. Therefore, in order to solve the above limitations, we propose the following strategies.

#### 3.3.1. Add a Weight Function to the Loss Function

One way to improve the accuracy of PINNs is to add a weight function before various losses of the loss function. For small-parameter problems, the weight function can be increased to balance all kinds of losses, so that the network will not focus on training one item and ignore other items. For the problem we are trying to solve, according to the specificity of our loss function, we only add the weight function to $L(x_{fs},\theta )$ and $L(x_{fd},\theta )$, that is, we replace (Equation 7) with the following
(8)$$L\left(x,\theta \right)=\varphi \left(\nu \right)L\left(x_{fs},\theta \right)+L\left(x_{us},\theta \right)+\psi \left(\nu ,K\right)L\left(x_{fd},\theta \right)+L\left(x_{ud},\theta \right)+L\left(x_{u\Gamma },\theta \right),$$
where φ(ν) and ψ(ν,K) take the reciprocal of the corresponding parameters in the equation, that is, $\varphi \left(\nu \right)=\frac{1}{\nu }$, $\psi \left(\nu ,K\right)=\frac{K}{\nu }$. Here, we do not use some adaptive weighting strategies [39,40] because the purpose of adaptive weighting strategies is to accelerate the convergence of the loss function. By adjusting the weights of various losses in the loss function, the value of the loss function decreases rapidly, but this has little effect on the small parameter problem we want to study.

#### 3.3.2. Parallel Network Architecture

Another way to improve the approximation ability of PINNs is to decompose the solution region, that is, to divide the solution region into several sub-regions and use independent networks within each sub-region, which is the parallel network structure strategy. Common parallel network algorithms are conservative PINNs (cPINNs) [41] and XPINNs [16], both of which have been given parallel implementations in [42]. cPINNs are required to satisfy nonlinear conservation laws, and the interface condition part of the loss function is relatively complex, but XPINNs are suitable for solving all differential equations, and the interface condition part is also relatively simple. In the model to be solved, we use the idea of XPINNs to divide the solution region into two regions and train the neural network in the two sub-regions respectively. The specific training process is shown in Figure 2. The parallel network architecture has a very good effect on the solution of discontinuous problems on the interface, which will be shown in the numerical examples that follow.

#### 3.3.3. Local Adaptive Activation Function Strategy

The selection of activation function is very important for the training of neural networks. The use of a single activation function can no longer meet the needs of solving complex problems. Therefore, Jagtap et al. proposed Rowdy activation function [43] with good properties for solving partial differential equations with high-frequency composite components and proposed adaptive activation function. In [44], an additional scalable parameter na is introduced to the activation function, where n≥1 is a predefined scaling factor and parameter a∈R is the slope of the activation function. Since parameter *a* is defined for the whole network, we call this the global adaptive activation function (GAAF). The neural network expression of GAAF is shown by
(9)$$U_{\hat{\theta }}\left(z\right)=\left(N_{D}\circ \sigma \circ naN_{D-1}\circ \sigma \circ naN_{D-2}\circ \cdots \circ \sigma \circ naN_{1}\right)\left(z\right).$$

The optimization of these parameters will dynamically change the value of the loss function so as to accelerate the convergence of the loss function. But GAAF may fail on some complex issues. Therefore, a layer-wise locally defined activation function is proposed to extend this strategy, that is, add different slope *a* to the activation function of each hidden layer of the neural network. The neural network expression of the hierarchical local adaptive activation function is shown as follows:(10)$$U_{\hat{\theta }}\left(z\right)=\left(N_{D}\circ \sigma \circ na^{D-1}N_{D-1}\circ \sigma \circ na^{D-2}N_{D-2}\circ \cdots \circ \sigma \circ na^{1}N_{1}\right)\left(z\right).$$

This provides additional D−1 parameters and optimizes the weight and bias, i.e., $\hat{\theta }={\left\{W_{i},b_{i},a_{i}\right\}}_{i=1,2,\cdots ,D-1}\in \hat{\Theta }$. Here, unlike the global adaptive activation function, each hidden layer has its own activation function slope.

## 4. Numerical Experiments

This section introduces several numerical experiments to solve the two-dimensional coupled Stokes–Darcy equation. Firstly, the accuracy and validity of the numerical method are verified by constructing numerical examples with analytical solutions, and the influence of weighted loss function on solving small parameter physical problems is demonstrated. Then, the analytical solution of interface discontinuity is constructed, and the numerical results of two different network structures are compared. Then, the more complicated interface curve problem is solved. Finally, a numerical example without analytical solution is designed to simulate the fluid movement under different permeabilities and viscosities, and the velocity flow diagram, in accordance with the physical law, is obtained.

In the following examples, we use the relative $L_{2}$ norm to estimate our error by
(11)$$E=\frac{\sqrt{\sum_{i=1}^{i=N}{\left|U_{exact}^{i}-U_{pred}^{i}\right|}^{2}}}{\sqrt{\sum_{i=1}^{i=N}{\left|U_{exact}^{i}\right|}^{2}}}.$$

Here, N represents the number of all points in the neural network training process, $U_{pred}$ represents the predicted value at the corresponding coordinate point, and $U_{exact}$ represents the analytical value at the corresponding coordinate point.

### 4.1. Interface Continuous Solution Problem

It is difficult to find a solution that meets the interface conditions ([Disp-formula FD4b-entropy-24-01106]) and ([Disp-formula FD4c-entropy-24-01106]). In this case, we simply extend the interface conditions to include an inhomogeneous term based on benchmark problem in [30,31]. In other words, we replace ([Disp-formula FD4b-entropy-24-01106]) and ([Disp-formula FD4c-entropy-24-01106]) with
(12a)$$\begin{array}{cccc} 2\nu n_{s}\cdot D\left(u_{s}\right)\cdot n_{s}= & p_{s}-p_{d}+g_{1},\; & \mathrm{on} & \;\Gamma , \end{array}$$
(12b)$$\begin{array}{cccc} 2n_{s}\cdot D\left(u_{s}\right)\cdot \tau = & -\alpha K^{-1/2}u_{s}\cdot \tau +g_{2},\; & \mathrm{on} & \;\Gamma . \end{array}$$

Then, we consider the coupled Stokes–Darcy equation in the region Ω=[0,1]×[−1,1]. The interface is Γ=[0,1]×{0}, and set α=1; then, the analytical solution is given by
(13)$$\begin{array}{cc} u_{s}= & [-\sin{\left(\pi x\right)}^{2}\sin\left(\pi y\right)\cos\left(\pi y\right),\;\;\sin\left(\pi x\right)\cos\left(\pi x\right)\sin{\left(\pi y\right)}^{2}],\\ p_{s}= & \sin(\pi x)\cos(\pi y),\\ u_{d}= & [-\sin{\left(\pi x\right)}^{2}\sin\left(\pi y\right)\cos\left(\pi y\right),\;\;\sin\left(\pi x\right)\cos\left(\pi x\right)\sin{\left(\pi y\right)}^{2}],\\ p_{d}= & \sin(\pi x)\cos(\pi y). \end{array}$$

Figure 3 shows the comparison between the analytical solution and the neural network prediction solution. Through (Equation 11), the relative $L_{2}$ error of each physical quantity can be calculated as $E(u_{s})$ = $4.04\times 10^{-4}$, $E(u_{d})$ = $1.46\times 10^{-3}$, $E(p_{s})$ = $3.48\times 10^{-3}$ and $E(p_{d})$ = $4.22\times 10^{-5}$, respectively. In the Table 1, we give the hyper-parameters in the neural network training process, where $N_{us}$ and $N_{ud}$ represent the number of points taken on the border of the Stokes region and Darcy region; $N_{fs}=N_{fd}$ and $N_{fs}=N_{fd}$ represent the number of internal points; $N_{u\Gamma }$ represents the number of interface points; $L_{N}$ and $N_{N}$ represent the number of neural network layers and the number of neurons in each hidden layer; and $N_{P}$ represents the number of parameters of the neural network. The optimization algorithm, learning rate η, and activation function will continue to be used in the following examples unless otherwise specified. Next, in Table 2 and Table 3, when we set the permeability as $K=10^{-2}I$ and $10^{-4}I$, respectively, and the fluid viscosity as $\nu =10^{-1}$, $10^{-2}$, $10^{-3}$ and $10^{-4}$ respectively, calculating the relative error of each physical quantity. The results show that the weighting of loss function is more helpful to calculate small parameter problems. At the same time, when permeability $K=10^{-2}I$ and fluid viscosity $\nu =10^{-3}$, the change of loss value and the change of relative $L_{2}$ error of each physical quantity in the training process are shown in Figure 4. In the figure, each physical quantity with W in front represents the weighted result of the loss function, and the one without W in front represents the unweighted result of the loss function, which further shows that our measures are effective.

Next, we study the influence of the depth and width of the neural network on the prediction accuracy. In this study, we control other hyperparametric variables to remain unchanged. For different network depths and widths, the training times of Adam and L-BFGS algorithms are 10,000. As shown in Table 4 and Table 5, we observe that the prediction accuracy of the model will increase with the increase of the width and depth of the neural network.

### 4.2. Interface Discontinuity Solution Problem

In this example, we solve the Stokes–Darcy equation for discontinuous interfaces. Since the fluid is not continuous when passing through the interface, the solutions of the two regions will be very different, and it will be difficult to optimize, so it is difficult to simulate the fluid in the entire region with only one network. Therefore, we propose the parallel network architecture; one network is in the Stokes region, and the other network is in the Darcy region, and both networks play a role in the simulation at the interface. The solution region and parameter α are the same as in the previous example. The terms on the right-hand side of the equation and the inhomogeneous terms in the interface conditions are given by
(14)$$\begin{array}{cc} u_{s}= & [-\sin{\left(\pi x\right)}^{2}\sin\left(\pi y\right)\cos\left(\pi y\right),\;\;\sin\left(\pi x\right)\cos\left(\pi x\right)\sin{\left(\pi y\right)}^{2}],\\ p_{s}= & \frac{1}{2}\sin\left(\pi x\right)\cos\left(\pi y\right),\\ u_{d}= & [\frac{1}{2}\sin\left(2\pi x\right)\cos\left(2\pi y\right),\;\;-\frac{1}{2}\cos\left(2\pi x\right)\sin\left(2\pi y\right)],\\ p_{d}= & \frac{1}{2}\sin\left(\pi x\right)\cos\left(\pi y\right). \end{array}$$

Table 6 shows the comparison of the CPU-time used by the three algorithms to solve the model under the control of relevant variables, as well as the comparison of the relative $L_{2}$ error of each physical variable. It can be observed that the parallel network architecture takes less time and has less error. Figure 5 shows the prediction results of velocity variables in the model by three algorithms, i.e., the single network structure, parallel network structure, and parallel network structure, with a local adaptive activation function strategy, as well as the absolute error comparison of the three algorithms. It can be observed from the absolute error diagram that the single network structure has a significant impact on the vicinity of the interface when solving the model. The simulation is not very good, but the parallel network architecture can be well simulated at the interface. It should be noted that the comparisons in Figure 5 and Table 6 are based on the same premise of controlling other training parameters.

### 4.3. Curved Interface Problem

In this example, we solve the curve interface problem, solve the region Ω=[0,1]×[−1,1], interface Γ:y=0.0625sin(4πx), and make the parameter α=1. Since the interface is a curve, the outer normal vector n and tangent vector τ at each point on the interface are changed, so we make φ(x,y)=0.0625sin(4πx)−y; therefore, the outer normal vector at the interface is
$$n=\frac{\nabla \varphi }{\left|\nabla \varphi \right|},$$
here, we get $n_{s}=\frac{1}{\left|\nabla \varphi \right|}\left(\frac{d\varphi }{dx},\frac{d\varphi }{dy}\right)$, $n_{d}=\frac{1}{\left|\nabla \varphi \right|}\left(-\frac{d\varphi }{dx},-\frac{d\varphi }{dy}\right)$.

The right end term of the equation and the non-homogeneous term in the interface condition are given by
(15)$$\begin{array}{cc} u_{s}= & [\frac{2}{\pi }\sin\pi y\cos\pi y\cos\left(x\right),\;\;\left(-2+\frac{1}{\pi^{2}}\sin{\left(\pi y\right)}^{2}\right)\sin\left(x\right)],\\ p_{s}= & \left(e^{y}-e^{-y}\right)\sin\left(x\right),\\ u_{d}= & [\frac{2}{\pi }\sin\pi y\cos\pi y\cos\left(x\right),\;\;\left(-2+\frac{1}{\pi^{2}}\sin{\left(\pi y\right)}^{2}\right)\sin\left(x\right)],\\ p_{d}= & \left(e^{y}-e^{-y}\right)\sin\left(x\right). \end{array}$$

Table 7 shows the hyper-parameters in the neural network training process. Figure 6 shows the comparison between the simulated fluid velocity and the fluid velocity given by the analytical solution and shows our calculation effect through the absolute error diagram. It can be seen that the error is relatively large only near the curve interface, and the simulation in other places is very good. Figure 7 represents the distribution of data points in each region used in the training process and the change of relative error of each physical quantity. Through (Equation 11), the relative $L_{2}$ error of each physical quantity can be calculated as $E(u_{s})$ = $3.81\times 10^{-3}$ and $E(u_{d})$ = $3.74\times 10^{-3}$, respectively.

### 4.4. No-True Solution Problem

In this example, the physical phenomenon described by the Stokes–Darcy system is examined. Let $f_{s}=0$ and $f_{d}=0$ in (Equation 1) and (Equation 2), the fluid viscosity ν=1, and the solution region Ω=[0,1]×[−1,1]. The boundary conditions of the two regions are shown in Figure 8.

Table 8 shows the hyper-parameters we used during training. We simulated different physical phenomena exhibited when a fixed permeability changes the viscosity of the fluid, as shown in Figure 9. From the simulation results, it can be seen that as the viscosity of the fluid decreases, the motion of the fluid becomes more intense, and the amount of fluid flowing through the interface and the speed will increase. At the same time, the physical phenomenon exhibited when the viscosity of the fixed fluid changes the permeability is simulated, as shown in Figure 10. From the simulation results, it can be observed that as the permeability decreases, the amount of fluid passing through the interface will decrease, and the velocity of the fluid that has passed through the interface will also decrease. The simulation effects shown in Figure 9 and Figure 10 conform to certain physical laws.

## 5. Conclusions

In this paper, based on the PINN algorithm, we propose several strategies to improve the accuracy to solve the more complex Stokes–Darcy model, and the effectiveness of our proposed strategy is well verified in the example of small parameters and discontinuous interface. These strategies are not only applicable to the Stokes–Darcy system but also have a certain reference for the solution of other more complex multiphysics coupled models. However, in the process of solving the training network, we did not obtain the convergence speed of the algorithm, and the research on the minimum and saddle point problems in the optimization problem is also very meaningful.

## Figures and Tables

**Figure 1 entropy-24-01106-f001:**
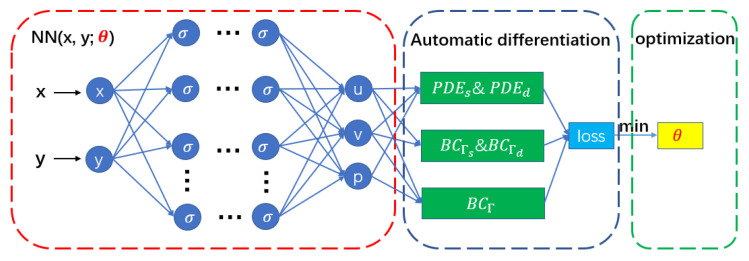
Physical-informed neural network structure diagram.

**Figure 2 entropy-24-01106-f002:**
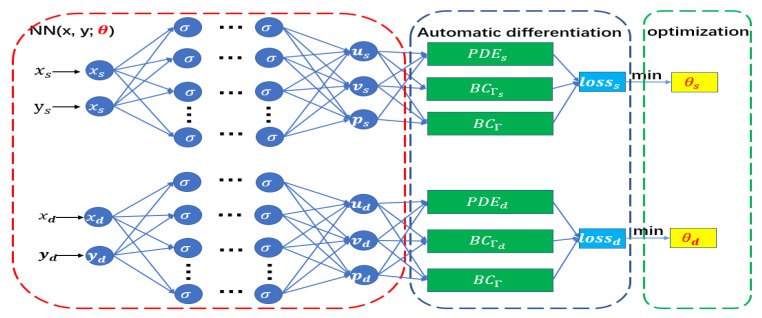
Parallel network architecture.

**Figure 3 entropy-24-01106-f003:**
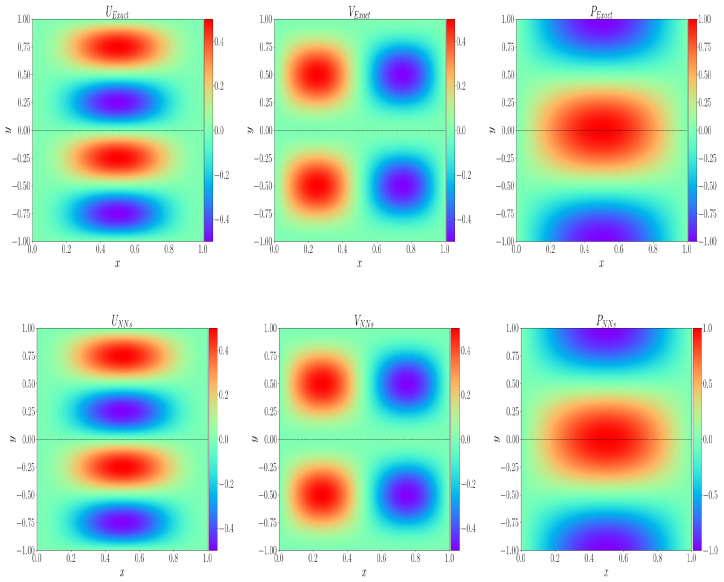
**Top**: Analytical solution of each solution variable. **Bottom**: The learning solution of each solution variable.

**Figure 4 entropy-24-01106-f004:**
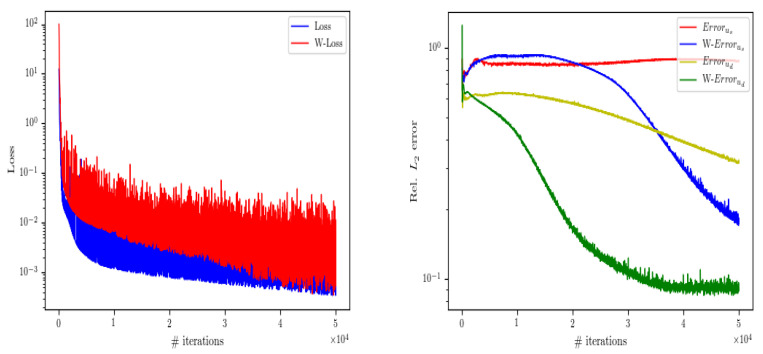
When permeability $K=10^{-2}I$ and viscosity $\nu =10^{-3}$, the change trend of weighted and unweighted loss function value (**left**) and the change trend of relative $L_{2}$ error of velocity (**right**).

**Figure 5 entropy-24-01106-f005:**
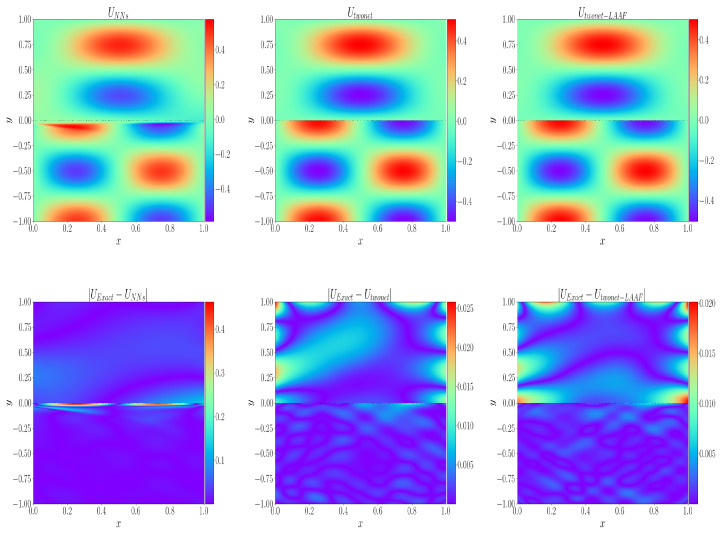
When the solution at the interface is discontinuous, the absolute errors of single network structure, parallel network structure, and local adaptive activation function parallel network structure are compared.

**Figure 6 entropy-24-01106-f006:**
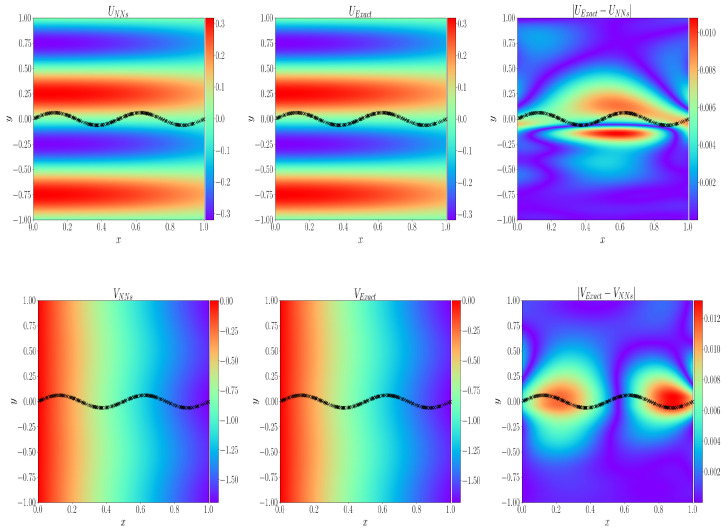
**Top**: Prediction solution, analytical solution, and absolute error of fluid velocity in x direction. **Bottom**: prediction solution, analytical solution, and absolute error of fluid velocity in y direction.

**Figure 7 entropy-24-01106-f007:**
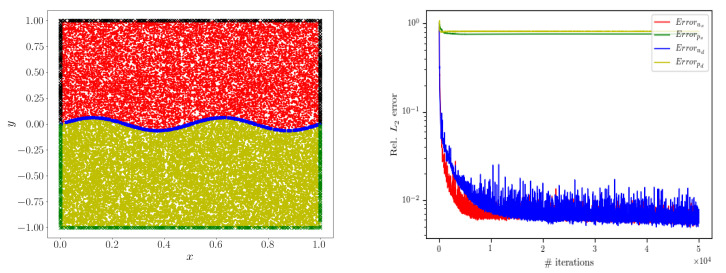
**Left**: the data points of each region are represented by different colors and symbols; **right**: the change of relative $L_{2}$ error of each physical quantity during the solution process.

**Figure 8 entropy-24-01106-f008:**
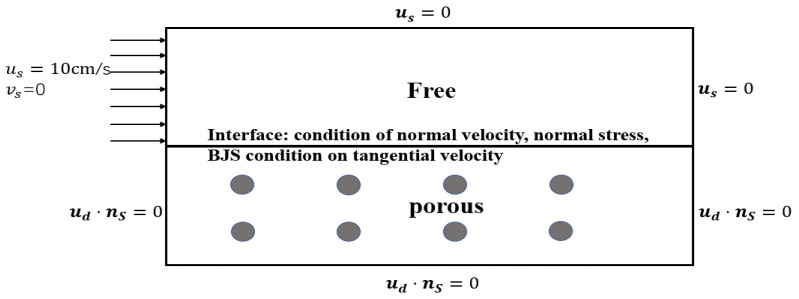
Computational domain and boundary conditions of Stokes–Darcy coupling model.

**Figure 9 entropy-24-01106-f009:**
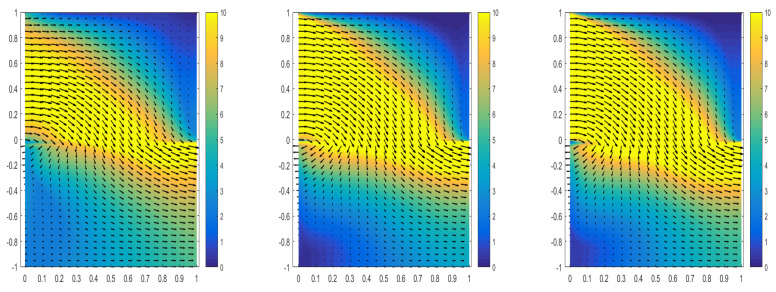
The figure shows that when the permeability of porous media is K=I, the change of fluid velocity is observed by changing fluid viscosity. **Left**: diagram of fluid velocity change when ν=1. **Middle**: diagram of flow velocity change when $\nu =10^{-1}$. **Right**: diagram of flow velocity change when $\nu =10^{-2}$.

**Figure 10 entropy-24-01106-f010:**
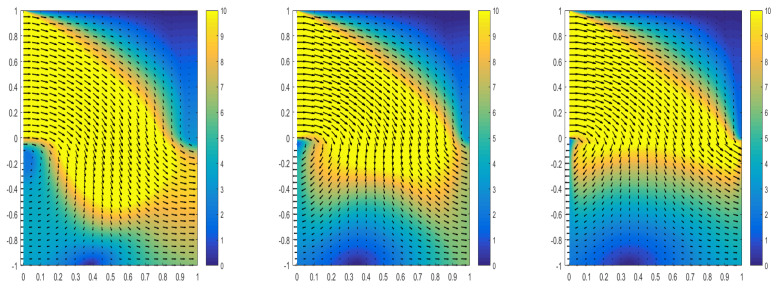
The figure shows that when the fluid viscosity is fixed at $10^{-4}$, the permeability of the porous medium is changed to observe the change of fluid velocity. **Left**: Graph of fluid velocity change when porous media permeability K=I. **Middle**: the change of fluid velocity when the permeability of porous medium $K=10^{-2}I$. **Right**: Variation diagram of fluid flow rate when porous media permeability $K=10^{-4}I$.

**Table 1 entropy-24-01106-t001:** Some hyper-parameters in the neural network training process.

$N_{\mathrm{us}}=N_{\mathrm{ud}}$	$N_{u\Gamma }$	$N_{\mathrm{fs}}=N_{\mathrm{fd}}$	$L_{N}$	$N_{N}$	OptAlgorithm	η	ActFunction	$N_{P}$
500	125	15,000	5	100	Adam&L-BFGS	$10^{-3}$	y=tanh(x)	30,903

**Table 2 entropy-24-01106-t002:** When $K=10^{-2}I$, the relative $L_{2}$ error of velocity and pressure under different fluid viscosities.

	φ(ν)=1,ψ(ν,K)=1				$\varphi \left(\nu \right)=\frac{1}{\nu }$, $\psi \left(\nu ,K\right)=\frac{K}{\nu }$			
ν	$E(u_{s})$	$E(u_{d})$	$E(p_{s})$	$E(p_{d})$	$E(u_{s})$	$E(u_{d})$	$E(p_{s})$	$E(p_{d})$
$10^{-1}$	$7.31\times 10^{-4}$	$8.91\times 10^{-4}$	$1.12\times 10^{-3}$	$5.34\times 10^{-4}$	$3.60\times 10^{-4}$	$4.58\times 10^{-4}$	$7.32\times 10^{-4}$	$2.57\times 10^{-4}$
$10^{-2}$	$9.19\times 10^{-4}$	$2.99\times 10^{-3}$	$1.61\times 10^{-4}$	$9.07\times 10^{-5}$	$6.37\times 10^{-4}$	$1.98\times 10^{-3}$	$9.71\times 10^{-5}$	$8.43\times 10^{-5}$
$10^{-3}$	$6.35\times 10^{-2}$	$1.78\times 10^{-2}$	$2.37\times 10^{-4}$	$6.85\times 10^{-5}$	$2.39\times 10^{-3}$	$8.64\times 10^{-3}$	$3.26\times 10^{-5}$	$3.31\times 10^{-5}$
$10^{-4}$	$9.13\times 10^{-1}$	$2.62\times 10^{-1}$	$9.73\times 10^{-5}$	$3.32\times 10^{-5}$	$2.54\times 10^{-2}$	$5.52\times 10^{-2}$	$1.51\times 10^{-5}$	$1.68\times 10^{-5}$

**Table 3 entropy-24-01106-t003:** When $K=10^{-4}I$, the relative $L_{2}$ error of velocity and pressure under different fluid viscosities.

	φ(ν)=1,ψ(ν,K)=1				$\varphi \left(\nu \right)=\frac{1}{\nu }$, $\psi \left(\nu ,K\right)=\frac{K}{\nu }$			
ν	$E(u_{s})$	$E(u_{d})$	$E(p_{s})$	$E(p_{d})$	$E(u_{s})$	$E(u_{d})$	$E(p_{s})$	$E(p_{d})$
$10^{-1}$	$9.48\times 10^{-2}$	$3.07\times 10^{-3}$	$6.32\times 10^{-2}$	$6.86\times 10^{-1}$	$1.05\times 10^{-2}$	$2.51\times 10^{-3}$	$1.35\times 10^{-2}$	$5.13\times 10^{-1}$
$10^{-2}$	$1.62\times 10^{-2}$	$5.41\times 10^{-3}$	$1.95\times 10^{-3}$	$6.97\times 10^{-2}$	$4.81\times 10^{-3}$	$2.90\times 10^{-3}$	$7.65\times 10^{-2}$	$3.81\times 10^{-2}$
$10^{-3}$	$8.15\times 10^{-1}$	$7.53\times 10^{-3}$	$1.54\times 10^{-3}$	$8.25\times 10^{-3}$	$5.31\times 10^{-3}$	$5.62\times 10^{-3}$	$6.53\times 10^{-5}$	$5.59\times 10^{-3}$
$10^{-4}$	$8.06\times 10^{-1}$	$5.24\times 10^{-2}$	$2.17\times 10^{-3}$	$2.23\times 10^{-3}$	$2.25\times 10^{-2}$	$1.89\times 10^{-2}$	$1.92\times 10^{-5}$	$1.00\times 10^{-3}$

**Table 4 entropy-24-01106-t004:** The influence of neural network width on the prediction accuracy of each physical variable.

	$E(u_{s})$	$E(u_{d})$	$E(p_{s})$	$E(p_{d})$
[2] + 4 × [10] + [3]	$2.71\times 10^{-2}$	$9.35\times 10^{-2}$	$2.52\times 10^{-1}$	$3.73\times 10^{-3}$
[2] + 4 × [20] + [3]	$6.88\times 10^{-3}$	$1.09\times 10^{-2}$	$3.58\times 10^{-2}$	$7.96\times 10^{-4}$
[2] + 4 × [40] + [3]	$3.23\times 10^{-3}$	$4.87\times 10^{-3}$	$2.01\times 10^{-2}$	$2.36\times 10^{-4}$
[2] + 4 × [60] + [3]	$1.21\times 10^{-3}$	$4.16\times 10^{-3}$	$1.58\times 10^{-2}$	$1.91\times 10^{-4}$
[2] + 4 × [80] + [3]	$1.00\times 10^{-3}$	$3.53\times 10^{-3}$	$1.03\times 10^{-2}$	$1.64\times 10^{-4}$

**Table 5 entropy-24-01106-t005:** The influence of neural network depth on the prediction accuracy of each physical variable.

	$E(u_{s})$	$E(u_{d})$	$E(p_{s})$	$E(p_{d})$
[2] + 2 × [60] + [3]	$3.50\times 10^{-3}$	$8.69\times 10^{-3}$	$1.23\times 10^{-2}$	$4.86\times 10^{-4}$
[2] + 4 × [60] + [3]	$1.45\times 10^{-3}$	$4.27\times 10^{-3}$	$2.19\times 10^{-2}$	$1.82\times 10^{-4}$
[2] + 6 × [60] + [3]	$1.46\times 10^{-3}$	$3.94\times 10^{-3}$	$2.16\times 10^{-2}$	$1.39\times 10^{-4}$
[2] + 8 × [60] + [3]	$1.09\times 10^{-3}$	$3.21\times 10^{-3}$	$1.06\times 10^{-2}$	$1.04\times 10^{-4}$

**Table 6 entropy-24-01106-t006:** The parameters of single network structure, parallel network structure, and parallel network structure using local adaptive activation function are compared.

	Single Network	Parallel Network, a = 1	Variable a, (n = 20)
network architecture	[2] + 4 × [100] + [3]	[2] + 4 × [70] + [3] (double)	[2] + 4 × [70] + [3] (double)
$N_{P}$	30,903	30,666	30,666
Training times	50,000	50,000	50,000
N	31,000	31,000	31,000
CPU-time(s)	11,482.7207	8482.6347	10,607.3143
$E(u_{s})$	$2.25\times 10^{-1}$	$4.28\times 10^{-2}$	$1.48\times 10^{-2}$
$E(u_{d})$	$3.41\times 10^{-1}$	$1.05\times 10^{-2}$	$3.66\times 10^{-3}$
$E(p_{s})$	$9.41\times 10^{-1}$	$1.75\times 10^{-1}$	$1.51\times 10^{-1}$
$E(p_{d})$	$1.51\times 10^{-2}$	$2.37\times 10^{-3}$	$1.17\times 10^{-3}$

**Table 7 entropy-24-01106-t007:** Some hyper-parameters in the neural network training process.

$N_{\mathrm{us}}=N_{\mathrm{ud}}$	$N_{u\Gamma }$	$N_{\mathrm{fs}}=N_{\mathrm{fd}}$	$L_{N}$	$N_{N}$	$N_{P}$
500	200	15,000	5	100	30,903

**Table 8 entropy-24-01106-t008:** Hyper-parameters in neural network training.

$N_{\mathrm{us}}=N_{\mathrm{ud}}$	$N_{u\Gamma }$	$N_{\mathrm{fs}}=N_{\mathrm{fd}}$	$L_{N}$	$N_{N}$	$N_{P}$
375	125	15,000	5	100	30,903

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
