# Peer review of "Physics-Informed Neural Networks for Solving Coupled Stokes–Darcy Equation"

_entropy, 2022, doi:10.3390/e24081106_

Round 1

Reviewer 1 Report

Please see the attached file containing my reviews.

Reviewer 2 Report

The authors considered using Physics-informed Neural Networks (PINNs) to solve the coupled Stokes-Darcy equation. The PINN method is a mostly developed deep learning method for solving differential equations. Several techniques including adding weight functions, parallel architectures and local activation functions were investigated to improve the performance of the PINN method. Some good results were obtained, and it may worth publishing. However, the PINN method is actually not a good option to solve the forward problem comparing with the traditional method. Maybe the authors can give some stronger comments on the motivation. More interesting work should be focused on the inverse problems by using PINNs. Also, the English should be improved.

Reviewer 3 Report

Report on “An improved physics-informed neural networks for solving coupled Stokes-Darcy equation (entropy-1831322)”  

by Ruilong Pu, Xinlong Feng and Hui Xu

In this article, the authors solving coupled Stokes-Darcy equations with Bever-Joseph-Saffman interface conditions by improve the physics-informed neural networks. The paper has some novelty and research value, and the experiments are relatively complete. There are some minor problems. I recommend a minor revision and acceptance after revision. The details are presented as follows:

1. Grammatical and writing errors in the full text need to be corrected well. (Eg. line 36 and etc.)

2. The English needs to be improved. There are many chinenglish expression in the paper, please check carefully and completely. (line 144: If you take... and many other sentences.)

3. n_s is defined, but not n_d (line 66).

4. What does “multi-empty medium region” means?(line 80)

5. Please explain the K in right hand of equation. (line 82)

6. What does z, N and sigma in equation (6) means? Please give the explanation.

7. What does the sentence in line 106 means?

8. “Then we find the solution of the partial differential equation” in line 131, please explain whether the solution is an exact solution or an approximated one.

9. Line 135, please ensure the completeness of the sentence.

10. Lines 137-139, the content is not consistent with the context.

11. Please explain how the improved PINNs to solve the  problem with discontinuous analytical solution on the interface in subsection 3.3.1. And why the weights function are chosen by taking the reciprocal of the corresponding parameters in the equation? Whats the advantage of taking the reciprocal of the parameters?  

12. In Figure 6, right one: please explain why the Error u_s and Error u_s stay in 10^0 and has no change with iteration increases? 

Round 2

Reviewer 1 Report

The authors answers all my concern satisfactorily. This manuscript can be accepted in the present form.

Reviewer 3 Report

I have no more comments.